# A Report on Smoking Detection and Quitting Technologies

**DOI:** 10.3390/ijerph17072614

**Published:** 2020-04-10

**Authors:** Alessandro Ortis, Pasquale Caponnetto, Riccardo Polosa, Salvatore Urso, Sebastiano Battiato

**Affiliations:** 1Department of Mathematics and Computer Science, University of Catania, Viale A. Doria, 6, 95125 Catania, Italy; sebastiano.battiato@unict.it; 2Center of Excellence for the Acceleration of Harm Reduction, University of Catania, Via Santa Sofia 89, 95123 Catania, Italy; p.caponnetto@unict.it (P.C.); polosa@unict.it (R.P.); toti.urso@eclatrbc.it (S.U.)

**Keywords:** smoking detection, health technologies, smoking cessation, medical mobile apps, technology review, wearable devices

## Abstract

Mobile health technologies are being developed for personal lifestyle and medical healthcare support, of which a growing number are designed to assist smokers to quit. The potential impact of these technologies in the fight against smoking addiction and on improving quitting rates must be systematically evaluated. The aim of this report is to identify and appraise the most promising smoking detection and quitting technologies (e.g., smartphone apps, wearable devices) supporting smoking reduction or quitting programs. We searched PubMed and Scopus databases (2008-2019) for studies on mobile health technologies developed to assist smokers to quit using a combination of Medical Subject Headings topics and free text terms. A Google search was also performed to retrieve the most relevant smartphone apps for quitting smoking, considering the average user’s rating and the ranking computed by the search engine algorithms. All included studies were evaluated using consolidated criteria for reporting qualitative research, such as applied methodologies and the performed evaluation protocol. Main outcome measures were usability and effectiveness of smoking detection and quitting technologies supporting smoking reduction or quitting programs. Our search identified 32 smoking detection and quitting technologies (12 smoking detection systems and 20 smoking quitting smartphone apps). Most of the existing apps for quitting smoking require the users to register every smoking event. Moreover, only a restricted group of them have been scientifically evaluated. The works supported by documented experimental evaluation show very high detection scores, however the experimental protocols usually lack in variability (e.g., only right-hand patients, not natural sequence of gestures) and have been conducted with limited numbers of patients as well as under constrained settings quite far from real-life use scenarios. Several recent scientific works show very promising results but, at the same time, present obstacles for the application on real-life daily scenarios.

## 1. Introduction

The last decade has observed a rapid growth and spread of several portable devices, which have taken a central role in all our daily activities. In particular, the smartphone technology, jointly with the availability of ever-increasing bandwidth connection and the growth of Social Networks, has changed the way people perform almost all their daily activities, as digital technology assumed a pervasive dimension. Beside smartphones, there has recently been a widespread use of several wearable devices or devices installed at home/office, which are connected to each other and are able to be controlled and actuated just by a simple smartphone app. Such a device ecosystem is aimed to make our devices and environments more “smart” (e.g., smart homes, smart offices, etc.). Indeed, specific sensors included in both wearable and remote devices can record specific parameters related to the monitored person or environment and share such data with the other devices. The data coming from different devices can be further processed and analysed jointly, with the aim to get insights and create value for the user experience. This provides opportunities for both capturing data and supporting users. A new innovative application of smart devices technology is the support for smoking cessation treatments.

In this paper, we present and discuss the state-of-the-art in technology focusing on automatic smoking detection and smoking quitting systems. An automatic smoking detection system is a technological solution designed to infer the number of cigarettes that have been smoked by a subject within a period of observation. This definition includes approaches that require the minimal intervention by the user (i.e., automatic), handling all the steps involved toward the detection of smoking events, from the extraction of sensory data to the final inference, the opposite of solutions that rely on participant self-report (i.e., diary apps). In particular, this paper presents and compares the different smartphone applications (i.e., apps), wearable technologies for automatic smoking detection, and all the other cases in which technology can support smoking cessation interventions. In the following sections, the most relevant smartphone apps and systems designed to help people stop smoking are presented and compared. A summary of the revised solutions to help users quit smoking is presented in the final Section.

## 2. Materials and Methods 

For the purposes of this research, we considered a first group of restricted papers detected by means of an exhaustive electronic search for relevant literature. Then, pertinent articles were exploited as seed papers to extend the selected papers. In particular, we conducted a search of recent papers using PubMed and Scopus databases with the terms “smoking detection system” and “sensors for smoking detection”, considering papers published in the last ten years. This first research returned a total of 31 unique documents. References from the resulting articles have been further analysed for relevant content, considering the papers’ aims, the employed devices, key results and insights. Since the use of wearable devices for smoking detection is rather new, most of the retrieved works are related to products currently under evaluation, or still at an experimental prototype stage. The smoking quitting apps were selected as follows: First, a Google search with the term “quit smoking apps” was performed. Then, from the first page of results, the entries related to sponsored links promoting a specific app were excluded. The resulting links were related to a list of blogs dedicated to health topics (e.g., healtline.com). Then, from the resulting list of apps, we selected the apps that report a high average feedback rating provided by the users in the Android and iOS app stores (i.e., rating 4/5 or higher). Apps that were not available in the English language were excluded. We categorized the reviewed technologies into two groups: smoking detection systems and smoking quitting apps. Within each distinct group, we also highlighted when a work was supported by scientific papers assessing its effectiveness or has been approved by a scientific advisory committee, which have been included in the references.

## 3. Smoking Detection Technologies

In this paragraph, we describe a group of technologies aimed to detect smoking events. The most clear and important benefits from these approaches is that the user is not required to log each smoked cigarette or smoking event, and, therefore, an objective reporting of smoking events is provided in real-time. Some of these technologies are still under evaluation, others are available on the market. Most of these technologies are based on the interaction between the smartphone app and a wearable device, typically an armband or a smartwatch, aimed to detect the smoking events by the algorithmic analysis of special movements performed by the user (i.e., hand-to-mouth motion). The first research attempts to investigate the detection of smoking events were carried out by Lopez-Meyer [1], in which the authors present a prototype system for automatic smoking detection. The system includes different sensors able to capture hand-to-mouth gestures and the tidal volume during respiration. The patient wears a radio frequency (RF) sensor on the wrist, and a receiver (i.e., antenna) on the chest. The receiver detects a signal whose strength is proportional to its distance to the RF sensor. Therefore, the antenna records different signal strengths, related to the distance from the RF on the user’s wrist. At the same time, a commercial Respiratory Inductive Plethysmograph (zRIP) captures the change in the volume of the user’s lungs, by means of thoracic (TC) and abdominal (AB) elastic sensor bands (see Figure 1). The sensor signals are analysed by a Support Vector Machine (SVM) classifier (see the Glossary) to detect smoking actions. Twenty regular smokers (i.e., carbon monoxide breath sample > 10 ppm) were selected, 10 males and 10 females, aged between 19 and 27 (see Table 1). The participants were asked to perform 12 different activities, including smoke a cigarette while sitting or standing. In particular, the selected activities were:Sitting comfortablyReading aloudStanding stillWalk on a treadmill in a self-selected slow paceWalk on a treadmill in a self-selected fast paceUse a computer to browse the InternetEat food using the hands and drink from a cupEat food using silverware and drink using a strawWalk outside the laboratory buildingSmoke a cigarette while sittingRest in sitting positionSmoke a cigarette while standing

The signals were stored on a microSD card flash memory and then processed offline. The smoking detection rate (i.e., recall) was 80%. The recall is the ratio between the number of correctly detected smoking events (true positives) and the total number of smoking events within a session (total positives). Hence, this metric quantifies the percentage of success on detecting the smoking events (see Appendix A—Classification Quality Measures). Although the work in Lopez-Meyer [1] implemented just a prototype, the achieved results represent the first step towards the implementation of an automatic smoking detection system.

The work in Patil [2] performed the same experiments proposed in Lopez-Meyer [1], by focusing on the analysis of the factors that affect the output quality of the Respiratory Inductive Plethysmograph (RIP) sensor. These factors include anthropometric variables such as gender, body mass index (BMI), dominant hand and the posture during smoking (sitting or standing). Indeed, subjects with different anthropometric characteristics (e.g., man and woman) may exhibit different types of breathing (abdominal or thoracic). The results of this study highlighted that the anthropometric characteristics of the person have a direct impact on the quality of the acquired signals and, hence, on the classifier performances. In particular, it was shown that posture (i.e., standing or sitting) and BMI (i.e., normal, overweight or obese) categories have a significant impact on the quality of breathing signals. Two different classification models have been developed. The individual models (which were subject-specific) achieved an average detection rate (i.e., recall) of 68%, whereas the group model (which was subject-independent) achieved an average recall of 61%.

The work in Lopez-Meyer [3] presents the design and the evaluation of a system based on a proximity sensor that captures hand-to-mouth gestures and detects smoking events, similar to the one described in Reference [1]. Differently than the previous work, this system does not need to capture the tidal volume during respiration. Figure 2 shows an example of the proposed system [3]. The data collection involved 20 participants, 10 males and 10 females, aged between 19 and 27. The volunteers were considered regular smokers based on the carbon monoxide breath measure higher than 10 ppm sampled at the time of the experiments. The participants performed the same 12 daily activities defined in Reference [1]. The smoking gesture has been detected performing specific thresholding on time and amplitude values: when the signal amplitude is higher than a pre-defined value, a hand-to-mouth (HTM) gesture is detected, then, gestures with duration atypical to smoking-related gestures are filtered. The experiments showed a smoking detection rate (i.e., recall) of 90% (see Appendix A—Classification Quality Measures).

The work described by Cole et al. [4] represents one of the first steps towards the development of methods by employing a commercial smartwatch inertial sensory data instead of a prototypal system of sensors, with the end-goal of integrating the proposed method in a novel real-time intervention for smoking cessation. It presents a Machine Learning system based on the use of an Artificial Neural Network (ANN) able to detect smoking events using only the accelerometer data collected via a smartwatch. This evaluation is based on an exploratory work, which demonstrated that the 3-axis accelerometer signal is enough to identify smoking gestures presented in Cole [5]. In particular, the x-axis is useful to detect smoking actions, whereas the y- and z-axes are helpful to eliminate other hand-to-mouth actions that may closely mimic that of smoking (e.g., eating, drinking and scratch of nose). The authors collected 120 h of accelerometer data using a smartphone and a smartwatch. Participants were provided with an Asus Zenwatch and an Android mobile. Participants were engaged in various daily activities, performing a self-reporting of smoking events. The true positive detection rate (i.e., recall) was 89%, whereas the false positive rate was 2.1% (see Appendix A—Classification Quality Measures). 

Detection of smoking events can also be identified by analysing 3-axis gyroscope raw signals that are obtained by any Android Wear and sent to a smartphone, as in Maramis [6]. A set of features extracted from the raw signals has been properly designed while the experiments were focused on the assessment of the discriminability potential of the features.

The authors of Shoaib [7] present SmokeSense, an online activity recognition system that logs data from various sensors and runs the activity recognition process in real-time and directly on the sensory devices (i.e., Android smartwatch). The performances on smoking detection have been evaluated under several variants of smoking (such as sitting, standing, walking, biking and other similar activities performed while smoking). Although this technology achieved a smoking detection rate of 89%, the experiments involved a limited number of participants (i.e., 11 participants for the general classifier evaluation and only 1 participant for the subject-specific evaluation). Therefore, the obtained results can be considered as just indicative (see Table 1).

The Case Western Reserve University (CWRU) developed a smoking cessation system able to perform the detection of smoking activities in real-time, based on the analysis of the movements registered by two armbands [8]. The system has three main components: the software application component, which consists of the quitting plan and mindfulness training, the activity recognition component, composed by the armbands’ raw signal data collection and the classification algorithm, and the Internet service component, that includes the message service and cloud sharing. In this paragraph, we focus on the second component, aimed to perform the detection of smoking activities, whereas the other two components will be described in the section concerning the smoking cessation tools. The activity recognition algorithm is based on a Long-Short-Term Memory (LSTM) neural network (see the Glossary). An LSTM is an artificial neural network specifically designed to analyse sequential data and time series. In this case, it is exploited to analyse the sequence of gestures of the user transmitted by the two armbands. In particular, the signal is segmented considering a pre-defined sequence length. Then, the features extracted from the sequential raw signal are the mean value, the standard deviation, the maximum value, the minimum value, the interquartile range and mean absolute deviation. The sequence of such features is then used as input to train the LSTM neural network. The experiments performed in the paper take into account six motions similar to smoking activity, such as answering the phone, drinking coffee, drinking water, sitting, standing and walking. The evaluation results are very interesting, as they show that the smoking activity recall (i.e., capability to correctly detect smoking motions) was 98.5%, the smoking activity precision was 76.8%, while the overall accuracy of the 6 motions was 72.6% (see Appendix A—Classification Quality Measures). Despite the impressive results, the experiments have been conducted in a very constrained way. Indeed, each activity was performed by applying a specific sequence of sub-actions for each hand. For example, the protocol for the activity “smoking” is the following: Both hands on the desk,Right hand: light the cigarette, left hand: block the wind,Right arm: raise near the mouth and inhale once and then put down, left arm: put down.

Such a structured and constrained way to record the signals associated to activities affects the interpretation of the very high achieved performances.

CigFree is a smart band, developed by iMorph, that detects the gesture of smoking and alerts the user with messages aimed to remind him all the reasons why he wants to quit smoking in the form of customised warnings (e.g., for your health, for your skin, for your friends, etc.). This device sends to the user, in his own voice, a personal message previously recorded (i.e., when the user is not affected by the smoking addiction). The company iMorph will conduct an IRB (Institutional Review Board)-approved open-label design study for 30 days to assess the extent to which CigFree use leads to a cessation or reduction of smoking [9]. A group of 100 people will be randomly selected and receive the CigFree cessation program.

SmokeBeat is a monitoring app developed by Somatix that exploits the movements of the hand to detect smoking events by analysing accelerometer and gyroscope signals transmitted by a smartwatch or a smart band. The SmokeBeat app is installed in both the smartphone and smartwatch. The watch transmits the gesture signals to the app in the smartphone that performs a detection of a sequence of puff events by exploiting Machine Learning techniques. When a smoke event is detected, it is notified on both devices, and the user can confirm or not confirm the correct detection. The algorithm sends the data to a remote system that refines the algorithm to fit the user habits and usual gesture while smoking. The app is available for both Android and iOS devices. A pilot study in the detection capability of SmokeBeat has been done by Dar [10]. This study involved 40 participants and reported a low false alarm rate, while the smoking detection rate (i.e., recall) was over 80% (see Table 1). Such a system, with higher detection rate, would be useful for diagnostic purposes, besides being used as a personal cessation tool by common users. Further developments of SmokeBeat technology, able to achieve higher performances, could allow for its application to clinical trials or epidemiology studies. For this reason, our research group performed a pilot study on the SmokeBeat smoking detection algorithm aimed to assess its eligibility for clinical trials. 

StopWatch is a smartwatch-based system for automatic detection of cigarette smoking presented by Skinner et al. [11]. Different from other methods, this system runs entirely on the smartwatch and does not require interactions with other sensors or a smartphone. In the pipeline proposed in Reference [11], the raw data (accelerometer and gyroscope) are first filtered and transformed in motion features. In particular, the raw signal values are first transformed by using statistical tools able to remove non-useful information, and then classified into one of the following motion features:Hand raising to mouthHand stationary at mouthHand moving away from mouth

Then, a decision tree classifier detects the drags from the sequence of motion features and their duration. Finally, an instance of smoking is associated to a sequence of drags, considering the number of consecutive drags and the time between drags. The evaluation was performed on 13 smokers in two phases: a laboratory session and real-life session (24 h). In the laboratory phase, the system obtained a precision of 75% and a recall of 92%, whereas in the free-living validation phase, the system obtained a precision of 86% and a recall of 71% (see Table 1). The system has been developed on an Android LG G-Watch.

The work in Reference [12] proposes an automatic smoking behaviour detector to provide real-time customized intervention messages to the smoker. The system exploits a wristband equipped with an accelerometer and a gyroscope. In the experiments, several settings have been evaluated, including accelerator-only, gyroscope-only and an accelerator–gyroscope approach. This allowed the authors to assess the contribution of each source of information. The evaluation data have been collected from five participants, aged between 30 and 35, performing hand-to-mouth gestures such as smoking, drinking, eating, scratching head and biting nails. The raw data have been collected by using an on-board 6-axis accelerometer and gyroscope from Arduino 101 development board, arranged on a wristband. The proposed approach extracts a pool of statistics from the raw data (e.g., min, max, average, interquartile distance, etc.) that are fed to an SVM (Support Vector Machine) or to a Random Forest classifier. The experiments showed that both orientation and rotation features are important for the aimed gesture classification task in terms of precision, recall and F1-score (see Appendix A—Classification Quality Measures). The best performances have been achieved by exploiting both orientation and rotation features and the Random Forest classifier approach, which obtained an average precision of 90%, average recall of 88% and an average F1-score of 88% (see Table 1). Although the results are interesting, the limited number and variety of the collected data do not allow for the assessment of the effectiveness of the method. Experiments should be performed on a more extended dataset, considering both the quantity and variety of participants, as well as more activities.

Senyurek et al. [13] exploited a wearable sensor system named PACT2.0 (Personal Automatic Cigarette Tracker 2.0) and proposed a method that combines information from an instrumented lighter and a 6-axis IMU (Inertial Measurement Unit) placed on the wrist for the detection of smoking events. An instrumented lighter captures the lighter press and release events, however no details of smoking duration can be found from this information. The paper presented a study performed on 35 moderate-to-heavy smokers in both controlled (1.5–2 h) and unconstrained free-living conditions (24 h). The collected information included about 871 h of IMU data, 463 lighting events and 443 cigarettes. The proposed method identifies smoking events from the cigarette lighter and estimates the puff counts by detecting proper hand-to-mouth gestures from the IMU data by exploiting a Support Vector Machine classifier. Experiments on controlled conditions achieved an average accuracy of 93% and an F1-score of 86% for the task of smoking event detection, whereas the scores achieved on the task of puff count estimation were 97% and 98%, respectively (see Table 1). Experiments in free-living conditions showed an 84.9% agreement with self-reports of participants. The high performances obtained in challenging settings with a consistent amount and variety of participants are encouraging. However, this study presents some limitations. Indeed, the proposed system considered the gestures performed only with the dominant hand (i.e., the method takes into account only one wrist device), whereas the addition of a second wrist device would allow for the detection of smoking with the non-dominant hand. Then, the PACT2.0 wrist device does not allow real-time streaming of the raw data. Indeed, the data can only be accessed after the acquisition session. This prevents the usage of the system in real-time scenarios.

Also, the work in Reference [14] exploited the PACT2.0 wrist module and the smart lighter used in Reference [13]. However, the approach in Reference [14] aimed to detect the smoking events by analysing the regularity of hand gestures. This was done by developing an unbiased autocorrelation process on the temporal sequence of hand gestures. Since the purpose of the study in Reference [14] was to investigate the contribution of the regularity and periodicity information for the task of smoking detection, in order to make the algorithm simple and computationally inexpensive, the detection was limited to a single axis of the inertial sensor. Experiments have been conducted both on controlled and free-living conditions. The authors collected 871 h of data, including 55 h in controlled conditions and 816 h of free-living recorded from 35 subjects (see Table 1). In the controlled experiments, the proposed approach achieved an accuracy of 98%, a recall of 97% and an F1-score of 81%. The combination of the smart lighter and the regularity of hand gestures allowed for reaching an F1-score of 91%, a recall of 92% and an accuracy of 83% in the free-living scenario.

The authors of Reference [15] proposed a mobile solution named RisQ, which leverages a 9-axis inertial sensor (accelerometer, gyroscope and a compass) mounted on a wristband, and an inference pipeline that combines a Random Forest classifier and a Conditional Random Field trained to detect smoking gestures in real-time (see Glossary). Rather than performing the classification directly on the raw information, the proposed system first extracts segments containing candidate gestures. Then, proper trajectory feature vectors are extracted from the segments. Sequences of trajectories are classified individually by a Random Forest, which also outputs the probability for the predicted gesture. The last step is aimed to classify the whole session based on the Random Forest outputs and a probabilistic Conditional Random Field that considers the co-occurrences of the different type of gestures in consecutive segments, previously classified by the Random Forest, and their probabilities. Experiments on smoking gesture detection reported an accuracy of 95.7%, precision of 91% and recall of 81%.

## 4. Smoking Quitting Applications

The apps described in this section are based on the annotation of smoking behaviours, achievements and craving events. These apps achieved very high average feedback ratings from users and high download rates in the app markets. In the following, we first present the apps that are supported by scientific third-party evaluation or approval, followed by the others.

The work presented by Chen et al. [8] also provides support for quitting smoking. In particular, a personalised quitting plan is created based on the answers provided by the user to several questions related to demographics and smoking habits (e.g., sex, age, smoking history, daily smoking quantity, etc.). The quitting plan is then defined, with the aim to maximise relevance to the individual, taking into account the provided answers. The user also specifies the goal, which will be to quit smoking or just reduce the number of cigarettes per day to a specific number. The app records and shows several information and statistics about the quitting plan and the smoking habits, such as the location of smoking. The software also suggests the number of cigarettes for each day, and this information is provided once the smoking gesture is detected by the detection component. The users periodically receive short videos which are proven to have positive effects for smoking cessation, motivational SMS (Short Message Service) as well as tips, suggestions and questions to check the user’s progress of smoking cessation. The quitting approach is based on the mindfulness exercise called the RAIN method [16]. RAIN stands for:Recognition of what you are experiencing,Acceptance of your emotions,Investigation of your thoughts and emotions,Non-identification with your thoughts and emotions, allowing them to pass naturally.

The four steps of RAIN are aimed to encourage the user to think carefully and try to manage his cravings [16]. The software application is developed for Android smartphones and is not on the market. Another component is the message service that is employed to send specific messages to the user periodically. The messages include quit plan tracking information such as triggered smoking events. The messages are sent to both the user and other people such as the doctors or family members, to inform them on the progress of the quit plan.

Pivot is a smoking cessation programme based on the use of a property breath sensor and a smartphone app. The Pivot breath sensor provides an immediate and objective measure of smoking effects to the user, indicating new insights related to his smoking behaviour. All the smoked cigarettes are registered by the user on the smartphone app, which will also store information of breath samples taken with the breath sensor. The dashboard of the Pivot app shows this information, together with other statistics such as the money spent on smoking. The app also delivers lessons, activities and interactions tailored to each patient’s needs and progress. Furthermore, the user receives personalised coaching support and feedback from an expert in smoking cessation. Pivot has been created by Carrot Inc., and has been approved by the U.S. Food and Drug Administration (FDA). The team that developed Pivot includes a large scientific advisory board of experts on medicine, heath education, psychiatry and behavioural sciences. It works on both iOS and Android smartphones (see Table 2). Pivot is available for companies aiming to provide their employees a quitting smoking programme. A demo can be requested from the website, then the individual employees can register and start their personal quitting program.

SmartStop is a smoking cessation system by Chrono Therapeutics. It consists of a nicotine replacement therapy designed to address smoking urges, especially in the morning. A wearable programmable patch is designed to deliver nicotine regularly, when the user has a peak of craving. The dosing patch communicates with the dedicated smartphone app and is programmed to deliver nicotine just before waking up. The nicotine is also delivered at midday and in the evening, when, statistically, a craving event happens. The smartphone app also provides periodic lessons on quitting smoking, tips, insights and suggestions to help the user to change daily habits properly. The smoking cessation programme has a duration of 10 weeks, in which the delivery of nicotine is optimised to avoid cravings. The development of this program, named CTI-100, is in phase 2 and is under examination of the FDA. The clinical trial involves 220 participants, and the study progress can be followed at the following URL: https://clinicaltrials.gov/ct2/show/NCT03178422.

Craving To Quit! proposes a 21 day quit programme developed and tested at Yale University [17]. It is possible to perform a three-day free trial of the App, then the cost is $24.99/month. There is also the possibility to have a lifetime membership at the cost of $139.99 (see Table 2). The users have access to videos, a cigarette consumption tracker and reminders. Moreover, for the premium users, a live coaching additional support is provided directly from the app. Cognitive behavioural therapy treatments are thought to act through the brain area involved in reasoning, planning and cognitive control in general (i.e., the prefrontal cortex). This is the part of the brain that helps us control the urge to pick up smoking. Like all the parts of the human body, the prefrontal cortex is subject to fatigue, also known as “ego depletion” [18]. When a person is hungry, angry, lonely or tired, they are more susceptible to smoking or using drugs, as this is the first brain region affected by stress conditions [19]. The quitting smoking programme of Craving to Quit! is based on mindfulness training, aimed to make the patient more aware of his smoking behaviour. Researchers have found that mindfulness training helps individuals with a range of addictions from alcohol to cocaine to nicotine dependence [20]. Indeed, the clinical trial presented in Reference [21] shows that individuals who receive mindfulness training as a stand-alone treatment for smoking cessation demonstrated greater reductions in smoking.

The effectiveness of mindfulness is related to the fact that it decouples the link between craving and subsequent smoking behaviour. An evaluation of such relationship between craving and cigarette use before and after individuals received mindfulness treatment for smoking cessation is presented by Elwafi et al. [22].

Clickotine is a clinically validated [23] digital smoking cessation programme which involves live coaching and nicotine replacement therapy. Clickotine has many features, such as: craving strategies like guided breathing, diary, community support, information about the health benefits and the money saved. The app is customisable on the user habits and goals. The programme includes access to an integrated data analytics portal that allows for tracking and monitoring the quitting progress. The evaluation study involved 416 participants for 8 weeks. After the 8-week study, 188 (45.2%) patients reported 7-day abstinence and 109 (26.2%) reported 30-day abstinence from smoking [23]. The programme is patent-pending. The base version of the app is free for both iOS and Android smartphones, options can be purchased (see Table 2).

2MorrowQuit is an app that implements a clinically proven smoking quit programme created in collaboration with the Fred Hutchinson Cancer Research Centre evaluated by Bricker et al. [24]. The programme is based on the use of Acceptance and Commitment Therapy (ACT) to help people change smoking habits. The programme starts by creating a personal user profile, used to define goals and prepare a quitting plan. When the user indicates a craving, the app proposes tips and suggestions, as well as personal live coaching. The study involved 99 people, among the patients that completed the programme (i.e., 24% of the total), the quit rates were 33% for 7-day point prevalence, 28% for 30-day point prevalence and 88% of participants just reduced their smoking frequency. The app is free and available for both iOS and Android devices, in-app purchases are available.

In the following, the smartphone apps for quitting smoking that are not supported by scientific assessment are briefly summarised (see Table 2).

LIVESTRONG MyQuit Coach is an app which acts as a virtual coach to develop a personalised plan for quitting. The user can choose the daily nicotine intake in order to quit smoking with a personalised timing. The app helps the user to keep track of the smoking consumption and nicotine cravings, shows the user his progress and motivating him to keep going with the programme. The app also defines goal-based achievements to keep the user motivated (e.g., the longest smoke-free streaks). Moreover, all the users are connected through the app built-in social circle, so that a user can receive support from other users. In this way, a user who has a craving day can be supported by the community. On the other hand, encouraging other users when they need help can be empowering. This app is available for both iOS and Android devices and is free (see Table 2).

Quit Smoking: Cessation Nation is an app for quitting smoking based on presenting motivational statistics and achievements and games aimed to distract the user when a craving event occurs. The app shows how much money the user is saving and body benefits from the smoking cessation activity. It also allows users to join each other to stay motivated through Facebook. This app is available only for Android devices and is free (see Table 2).

Quit Now! is a motivational app for quitting smoking. The app is available for both iOS and Android smartphones. The base version is free, but special options can be purchased to unlock additional features. The motivational messages lean on how much money the user is saving and the health benefits. Moreover, the user can chat with other participants who are quitting smoking directly via the app and get support. The app supports 44 languages. It is available for both iOS and Android devices and is free (see Table 2).

Quit Smoking with Andrew Johnson is an app for quitting smoking based on self-hypnosis to break the habit of smoking. The app mechanism is based on the listening of audio tracks designed by a clinical hypnotherapist (Andrew Johnson) that can help the patient to achieve a relaxed state. Although the app’s creators emphasise the exploitation of well-established relaxation techniques, there is not any documented experimental assessment associated to this product. This app is available for both iOS and Android devices at the cost of $2.99 (see Table 2).

Smoke Free is an app which offers a variety of tools to aid users to quit smoking. The dashboard shows how much money the user is saving from stopping smoking, the number of not-smoked cigarettes, health benefits as well as the number of cravings resisted and smoke-free days. The motivational system is based on progress badges that can be earned, sharing of achievements with friends and the record of cravings. Under the user’s explicit permission, the app will send all the recorded information that are exploited to further improve the app. The creators use the data information as part of a controlled trial done entirely via the app among a set of users selected randomly. When a new feature is developed, it is released to a random group of users and evaluated. The users are unaware if they are selected or not, but only the users who have accepted to take part in the experiments can be selected. The app’s creators claim that it is effective in helping people stop smoking, however this is an internal research that is not supported by published experiments (More details about how to join the trial are reported on the company’s website: http://smokefreeapp.com.) This app is available for both iOS and Android devices (see Table 2). The base version is free and additional features are available by purchasing.

Kwit is a motivational quitting smoking app based on the mechanism of gamification. Its aim is to augment the user’s participation, motivating him in the achievement of more than 60 scheduled goals. The user is motivated by emphasising personal achievements, providing health information related to the achieved goals, records and social media support. The app is supported by both iOS and Android based smartphones and is free (see Table 2).

Butt Out is a quitting smoking app based on the user’s logs of the cravings and smoking events to get data-driven insights to provide help to stay smoke-free over time. It is a diary-based app which shows health improvements, the money savings as well as personalised motivation photos to keep the person motivated. It is available for both iOS ($6.99) and Android smartphones ($2.99).

Get Rich or Die Smoking is a motivational quitting smoking app that uses money as motivation. It tracks the money the user saves by quitting and lets him know what he can afford once he has stopped smoking. It tracks health improvements, smoke-free days and other common statistics. This app is free and is available only for Android smartphones (see Table 2).

SmokeFree—Quit Smoking Slowly is a smoke quitting app that lets the user choose from two modalities of use: quit or reduce. Then, the app shows motivational images and videos promoting quitting in a personalised way. At the initialisation phase, the user claims the number of smoked cigarettes in a day, and the number of days to quit. Then, a personalised quitting plan is provided. The app keeps track of cravings and other statistics, moreover it allows the user to smoke only at certain periods and a limited number of times in a day, that is progressively reduced. The app is free and is available only for Android smartphones (see Table 2).

Quit Smoking NOW—Max Kirsten is a smoking cessation app sponsored by Max Kirsten, an affirmed smoking cessation expert and clinical therapist. The app keeps statistics, provides achievement reports, as well as a quit smoking timetable. Furthermore, it also helps the user curb cravings and avoid the usual weight gain. The app includes several tips defined by the expert, including his “60-Second Cravings Buster” technique to dissolve the cigarette cravings. It also includes hypnotherapy sessions, a quit smoking calculator, tips on “living life as a non-smoker”, a gallery of smoking-related images and support to stay away from smoking after quitting. The implemented techniques are based on hypnosis and neuro-linguistic programming (NLP). The app is available only for iOS smartphones for $4.99 (see Table 2).

Quit Tracker: Stop Smoking shows the money the user is saving and the health benefits of quitting. The diary is used to keep track of how the quitting is going. A graphic timeline shows the health improvements during the quitting process. The app is free and is available only for Android devices (see Table 2).

Quit It Lite is a motivational app that allows the user to define his/her personal smoking cessation goals, such as save money to buy a cappuccino, movie tickets or a new phone. It also provides a function to share the achievements with friends. The app is free and available only for iOS smartphones (see Table 2).

Quit Smoking Hypnosis is an app that provides daily sessions that claim to be effective within one week. The audio tracks are read by a certified hypnotherapist. The aim of the session is to induce the brain wave frequency into an optimal state for receiving hypnotic suggestions to quit smoking. However, the programme results are not clinically proven. The app is available for both iOS and Android smartphones. The base version is free and special features can be unlocked by payments.

QuitSTART is an app that takes the information provided by the user about smoking history and gives tailored tips, inspiration and challenges to help him quit smoking. The app monitors the progress and motivates the user to earn badges for smoke-free milestones and other achievements. It also helps to manage cravings and bad moods in healthy ways, distract the user from cravings with games and challenges, as well as provides a tool to share the progress through social media. The quitSTART app is a product of Smokefree.gov, a smoking cessation resource created by the Tobacco Control Research Branch at the National Cancer Institute. The app is free and available for both iOS and Android smartphones.

Quitter’s Circle is an app that aims to help the user to quit smoking with the support of friends and family, in addition to resources (e.g., tips, articles, etc.), a saving money calculator and progress tracking over time. It is possible to select a group of people to form a quit team named “inner circle”, aimed to support the user during the quitting process. By means of the quit fund tool, quitters can ask supporters for help covering any cost of quitting (e.g., treatments, doctor visits, etc.). It is possible to define a customisable quit plan, with goals and milestones (e.g., doctor visits). The app also works with the smartwatch to show notifications and summaries of the quitting progress. It is free and available for both iOS and Android smartphones.

## 5. Final Remarks

In this paper, we described the most relevant solutions for quitting smoking and detailed the related technology. Such solutions are based on the exploitation of smartphone technology, that is further supported by several recent scientific reviews such as Whittaker et al. [25] and Haskins et al. [26]. Imtiaz et al. [27] presented a review of wearable technology monitoring of cigarette smoking. In particular, the review presented in Reference [27] provides a systematic organisation and classification of the methods based on the exploited technology (e.g., inertial sensors, breathing sensors, acoustic sensors, cameras, etc.). The papers of the same category are compared by means of summary tables which report salient information, such as the type of the employed sensors and the amount of people involved in the validation study. However, the methods are not well detailed in the text, which provides only a brief description for each method. The paper in Reference [27] could be a useful guide for research scientists that are interested in technology for cigarette smoking monitoring, providing a structured classification of previous approaches, which can be used as a starting point for research in this field. Differently, this study presented and discussed the state-of-the-art on smoking detection technologies, focusing on the employed algorithms which are explained in detail, and on the related achieved experimental performances. Moreover, this paper presented different quitting smoking smartphone applications, with an emphasis on the apps that are supported by scientific third-party evaluation or approval.

In the first part, we described the most significant scientific works, in which wearable sensor technology has been exploited to define prototypal systems aimed to implement automatic smoking detection systems. Then, commercial products that implement the previous research insights were presented and compared. In the second part of the paper, we investigated current solutions and approaches for quitting smoking therapies, which are supported by the app and wearable technologies. These solutions include works that have been approved by a scientific committee, or are currently under evaluation, as well as smartphone applications available on the digital market that achieved high feedback ratings from users. All the described solutions are listed in Table 1 (smoking detection technologies) and Table 2 (smoking quitting apps).

One of the principal roles of many psycho-technological interventions is to change the behaviour of users. Antecedents—12, comparison of behaviour—16, covert learning—16, feedback on behaviour—2, goals and planning—1, natural consequences—5, regulation—11, reward and threat—10, scheduled consequences—14, self-monitoring—2.3 and social support—3, are the behaviour change techniques that are used in the quit smoking products described in Table 1 and Table 2. These behavioural interventions are coded according to the Behaviour Change Technique Taxonomy v1 (BCTTv1) [28]. Without this coding, researchers cannot describe the specific behavioural techniques they applicated in the technology used to stimulate the health empowerment. Several researchers or technology producers, for example, may declare that they utilised “motivational strategies” to encourage people to quit. Yet, each researcher or technology producer may have applied a different behavioural technique but used the same label.

Several recent scientific works show very promising results but, at the same time, present obstacles for the application in real-life daily scenarios. As an example, the work presented in Reference [8] shows impressive results. However, the evaluation protocol takes into account only six movements performed by only right-hand smokers, and the actions have been performed in a very constrained way, as previously detailed. Hence, it is not comparable with respect to the variability of a real case scenario, in which the user performs lots of daily activities in a very personal way. Furthermore, this system is based on two armbands, such a setting can be considered only in an experimental scenario, and this makes it difficult to transfer this scientific result to a real product. The works in Reference [13] and in Reference [14] achieved interesting performances in challenging experimental settings, which include a high number and variety of involved subjects, as well as several hours of free-living evaluation. However, the employed hardware (e.g., smart lighter) only allows for experimental evaluation that is far from a commercial solution. 

The evaluation protocols employed in the above-described scientific papers present several limitations: low number of participants/example data, low variability in the data collection (e.g., right/left-handed, male/female, etc.) and no comparison with respect to other methods in the state-of-the-art with the same dataset and evaluation protocol. Most of the described research works present the above issues. 

We plan to investigate the development of a fully comprehensive system, taking into account the most promising solution and properly assessing its suitability for a clinical programme.

While the majority of the apps rely on participant self-report (i.e., diary apps), solutions like SmokeBeat that exploit wearable sensors (i.e., armbands/smartwatch) have the potential to improve current approaches by providing automatic feedback and objective verification of smoking status. We believe that both research and industry in the field should invest effort and resources in this direction.

## 6. Conclusions

Although some of the described solutions are interesting, and supported by experimental results, most of them are business products with high performances from a technical point of view but are not reliable enough for a clinical program. For instance, in the case of wearable-based systems, anthropomorphic features such as the posture or the dominant hand may affect the performances. Moreover, the way by which a user wears the smartwatch/armband differs from person to person. The work in Reference [29] investigates the ambiguities in accelerometer data caused by the position of the sensors on a person, by considering eight common positions of a smartwatch on the wrist. The authors proposed a well-designed method that transforms the raw input data to a common representation, corresponding to an arbitrary configuration, and then recognises the gestures on the transformed data. Experiments have shown that the transformation improves the detection accuracy of a pre-trained predictor (i.e., an Artificial Neural Network) when it is evaluated with varying positions of the wearable sensors. Finally, the daily use of such technology includes a wide variety of actions, habits and gestures that are performed by the user in a very personalised way. Some people are used to smoking while performing other actions such as driving or doing manual working. In addition, any feasible solution that involves the usage of one or more battery-powered devices should also consider the concerns related to the limited power of such devices. In particular, the real-time signal processing algorithms should be designed to operate within the limited power available, with the aim to optimise the operational lifetime of the whole system. An example is given by the Bluetooth Low Energy (BLE) protocol [30], which represents a standard for data transfer for wearable devices aimed to be energy efficient. For these reasons, greater investments are needed in the study, development and in the scientific assessment of technology applied to smoking cessation with the aim to define reliable systems able to ensure high performances in real-life use scenarios.

## Figures and Tables

**Figure 1 ijerph-17-02614-f001:**
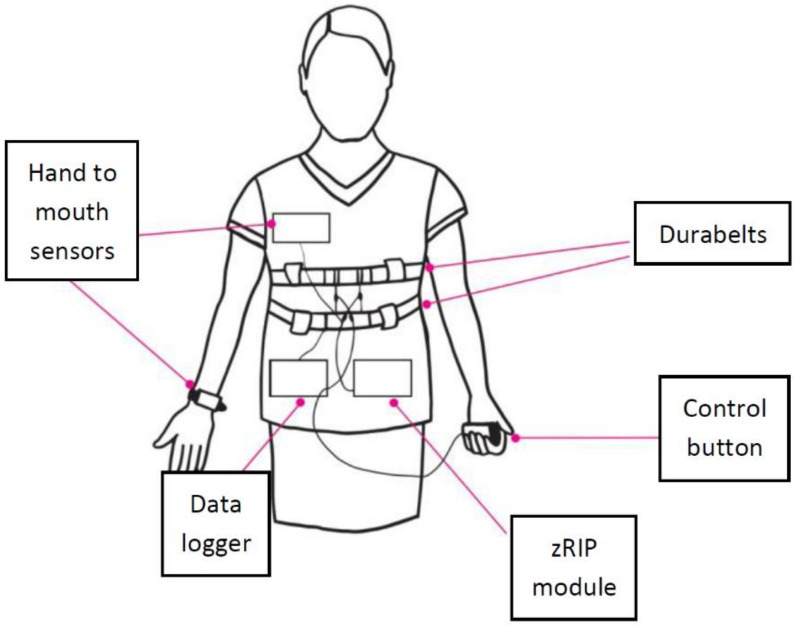
Sensors of the system presented in Reference [1].

**Figure 2 ijerph-17-02614-f002:**
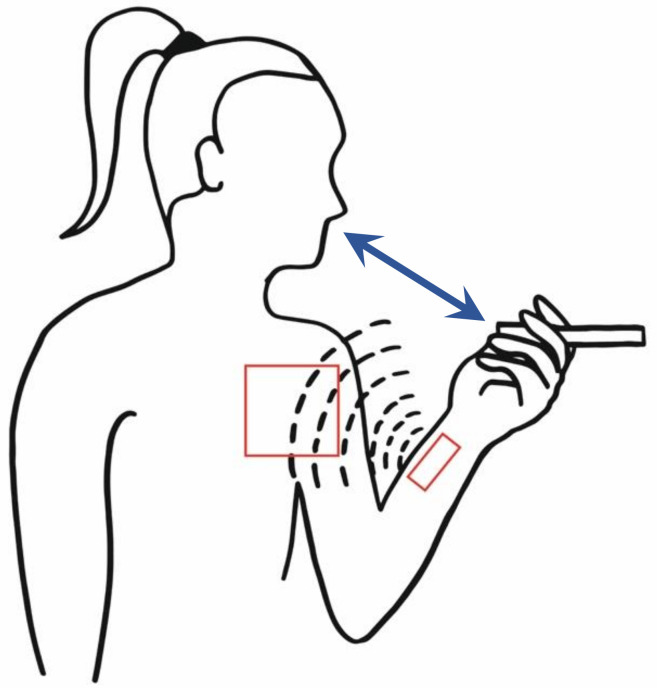
Schematic representation of the system showing the radio frequency (RF) transmitter and the receiver, as implemented in Reference [3].

**Table 1 ijerph-17-02614-t001:** Summary of the smoking detection technologies described in this study.

Product	Technology	iOS	Android	Scientific Evaluation	Price	Participants	Recall *
Lopez-Meyer et al. [1]	Hand-to-mouth and chest sensors			✔	Research Prototype	20	80%
Patil et al. [2]	Hand-to-mouth and chest sensors			✔	Research Prototype	20	61%
Lopez-Meyer et al. [3]	Hand-to-mouth sensor			✔	Research Prototype	20	90%
Cole et al. [4]	App + Smartwatch		✔	✔	Research Prototype	10	89%
Smokesense [7]	Smartwatch		✔	✔	Research Prototype	11	89%
Research by CWRU [8]	App + 2 Armbands		✔	✔	Not on the market	10	98%
CigFree [9]	App + Property Smartband			✔	Starting from $69	100	N/A
SmokeBeat [10]	App + Smartwatch	✔	✔	✔	Free	40	80%
StopWatch [11]	Smartwatch		✔	✔	Research Prototype	13	92%
Lu et al. [12]	Prototypal wrist sensors			✔	Research Prototype	5	88%
Senyurek et al. [13]	Prototypal wrist sensors			✔	Research Prototype	35	93%
Senyurek et al. [14]	Prototypal wrist sensors			✔	Research Prototype	35	97%
RisQ [15]	App + wrist sensors		✔	✔	Research Prototype	15	81%

* the recall values are related to the results reported by the related papers. However, since each paper applies its own evaluation protocol and each work performs the method evaluation on a different dataset, the recall values are not directly comparable to each other. The term “App” stands for smartphone application, CWRU stands for Case Western Reserve University.

**Table 2 ijerph-17-02614-t002:** Summary of the quit smoking applications described in this study.

Product	Technology	iOS	Android	Scientific Evaluation	Price
LIVESTRONG MyQuitCoach	App	✔	✔		Free
Quit Smoking: Cessation Nation	App		✔		Free
SmartStop	App + Smart Patch			✔	Product under development
Pivot	App + Property Breath Sensor	✔	✔	✔	Under request
Craving to Quit!	App	✔	✔	✔	Free (3 days) $24.99/month $139.99/life
QuitNow!	App	✔	✔		Free
Quit Smoking with A. Johnson	App	✔	✔		$2.99
Smoke Free	App	✔	✔		Free
Kwit	App	✔	✔		Free
Butt Out	App	✔	✔		$6.99 iPhone $2.99 Android
Get Rich or Die Smoking	App		✔		Free
Smoke Free Quit Smoking Slowly	App		✔		Free
Quit Smoking NOW—M. Kirsten	App	✔			$4.99
Quit Tracker: Stop Smoking	App		✔		Free
Quit It Lite	App	✔			Free
Quit Smoking Hypnosis	App	✔	✔		Free
quitSTART	App	✔	✔		Free
Quitter’s Circle	App	✔	✔		Free
Clickotine	App	✔	✔	✔	Free
2MorrowQuit	App	✔	✔	✔	Free

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
