# Peer review of "A Report on Smoking Detection and Quitting Technologies"

_ijerph, 2020, doi:10.3390/ijerph17072614_

Round 1

Reviewer 1 Report

In general, I find this to be a very complete and thorough survey of the existing technologies in this arena of research.  The following are my comments: 

Comments:

Lines 523-525: For instance, in the case of wearable based systems, anthropomorphic features such as the posture or the dominant hand may affect the performances. Moreover, the way by which a user wears the smartwatch/armband differs from person to person... 

These have been at least partially addressed in the following article: https://ieeexplore.ieee.org/abstract/document/7897312

In addition to the articles already reviewed you may also consider this article: https://www.ncbi.nlm.nih.gov/pmc/articles/PMC4682919/

Some grammatical issues. I have listed a few examples, but in general the paper should be read over with specific intent for correcting these types of errors. Examples:

  • Lines 42-44 "Beside smartphones, there has been a widespread use of several recently wearable devices or devices installed at home/office" -> recently available?
  • Line 52 "presents and compare" compare should be compares
  • ...
  • Lines 504-505

In Table 1 and Table 2 on page 12, why is iOS listed in Table 2, but not Table 1? For instance, SmokeBeat is available on iOS... 

Reviewer 2 Report

The authors of this manuscript present a very well aggregated and comprehensive review of the modern applications of mobile and wearable devices in the field of smoking. I highly appreciated the content of this review and I believe that many researchers will also strongly benefit from reviewing this manuscript. I therefore highly recommend its publications. I do however ask the authors to address the following two items:

1- The first item relates to the technical contents of this manuscript, which I found to be nearly complete. My only additional comment relates to the Final Remarks where the following statement was noted: 

"low variability in the data collection (e.g., right/left-handed, male/female, etc.)"

and 

"For instance, in the case of wearable based systems, anthropomorphic features such as the posture or the dominant hand may affect the performances. Moreover, the way by which a user wears the  Smartwatch/armband differs from person to person. Finally, the daily use of such technology includes a wide variety of actions, habits and gestures that
are performed by the user in a very personalized way."

Although I completely agree with the final conclusion of the authors that more work needs to be done, I do want to bring their attention to the following work that recognizes and addresses some of the noted limitations:

Resolving ambiguities in accelerometer data due to location of sensor on wrist in application to detection of smoking gesture

CA Cole, JF Thrasher, SM Strayer, H Valafar 2017 IEEE EMBS International Conference on Biomedical & Health Informatics …   In relation to the currently unexplored work, neither of the present reports discuss the pragmatic limitation of power and battery life, especially under the conditions of continuous use of the sensors. The authors may choose to review this aspect of wearables in the future.    2- My second main concern relates to the grammatical correctness of the manuscript. Although the main message of the manuscript is adequately conveyed, in its current form, I believe that the readability of the manuscript can improve through some grammatical corrections. These are too many, but simple to identify. I recommend using services such as Grammarly to review and correct these errors.   

Reviewer 3 Report

It is an interesting work which could be useful 1) for health professionals (not specialized in tobacco addiction) for advise smokers who want to stop without consulting a tobacco cessation practitioner, 2) for researchers for develop studies on automatic smoking detection systems, and for evaluate existing smoking cessation apps.

However, several elements have to be reviewed:

  • I think the following reference should be added: Imtiaz MH, Ramos-Garcia RI, Wattal S, Tiffany S, Sazonov E. Wearable Sensors for Monitoring of Cigarette Smoking in Free-Living: A Systematic Review. Sensors (Basel). 28 oct 2019;19(21).

  • Please precise what your work adds compared to this one (particularly for the part on smoking detection technologies)

  • Briefly define what an « automatic smoking detection system » is, in the introduction part (or give an example)

  • Develop much more the methodology part, for each work lead (i.e. smoking detection technologies and quitting apps): date (what is « recent papers »), is there limitations in the search algorithm, inclusion/exclusion criteria, study selection process, what is a « high average feedback ranting » (give numbers)…

  • For the result part: give a general description of results for each work lead, before giving the detailed results: number of technologies/apps identified during the search, number presented in the article, minimum & maximum of participants across all studies, minimum & maximum for the smoking detection rates, the download rates…

  • Cite tables 1 or 2 in the beginning of each paragraph of results

  • In the « smoking detection technologies » part:
    • When you first write about “recall” l105: explain what does this result mean in concrete terms
    • Figure 2 title: add ref [3], as you did for figure 1
    • L148 you give the false positive rate: can you give it for each product?
    • L 229: problem with the references order: go from [11] to [24]
    • L 240 what is « F1 score»? explain that

  • In the “smoking quitting applications” part:
    • Give a reference for each application
    • If available, give the main results for each application that are supported by scientific evaluation, as you did for Clickotine
    • L402 to 407: there is no published experiment, but can you give the reference where you find information about this trial?

  • Final remarks part
    • L 469 As the methodology is not detailed, we don’t understand how you can say that you describe “the most relevant solutions for smoke quitting” (idem l472)
    • L482-486: it is difficult to understand: first present the BCTT, and indicate how do you coded your results; this work should be developed
    • The difference among results in table 1 (61% for the minimum - 98% for the maximum) should be discussed
    • Please discuss the delimitations of your search: for the apps, did you take only English language apps? (for example, you don’t have the French app which has been (or maybe still in process) evaluated in a national randomized controlled trial - Cambon L, Bergman P, Le Faou A, Vincent I, Le Maitre B, Pasquereau A, et al. Study protocol for a pragmatic randomised controlled trial evaluating efficacy of a smoking cessation e-’Tabac Info Service’: ee-TIS trial. BMJ Open. 24 2017;7(2):e013604. )

  • Table 1:
    • The same technology should follow one after the other (e,g, StopWatch and SmokeSense)
    • What are references for SmokeBeat and CigFree?
    • Develop more this table: number of participants, other scores (false positive, etc)
    • You indicate that recall values are not comparable one each other: do you have data for creating a comparable score between the different products?

  • Table 2:
    • Order the apps as they appear in the main text (in the same order)
    • Add results for the apps which have been evaluated

  • Glossary: Maybe add some sentences about gyroscope, as you did it for the other systems

  • Some typing errors: l21 algothms, l99 staw, l124 doesn’t need do, l150 to a smartphone in Maramis, l153 the authors presents, l248 and proposes, l364 carving

Round 2

Reviewer 3 Report

Thanks for your corrections and clarifications; the manuscript is better.

I would have appreciated a reply about the BCT (Michie). However, as it is not your main objective in this work, the manuscript can be published in the present form.